# Preeclampsia and Future Implications on Growth and Body Composition in Preterm Infants

**DOI:** 10.3390/nu16213627

**Published:** 2024-10-25

**Authors:** Elisabeth Calek, Julia Binder, Pilar Palmrich, Felix Eibensteiner, Alexandra Thajer, Karin Harreiter, Angelika Berger, Christoph Binder

**Affiliations:** 1Department of Paediatrics and Adolescent Medicine, Division of Neonatology, Paediatric Intensive Care Medicine and Neuropaediatrics, Medical University of Vienna, 1090 Vienna, Austria; elisabeth.calek@meduniwien.ac.at (E.C.); alexandra.thajer@meduniwien.ac.at (A.T.); karin.harreiter@meduniwien.ac.at (K.H.); angelika.berger@meduniwien.ac.at (A.B.); 2Department of Obstetrics and Gynaecology, Medical University of Vienna, 1090 Vienna, Austria; julia.binder@meduniwien.ac.at (J.B.); pilar.palmrich@meduniwien.ac.at (P.P.); 3Department of Emergency Medicine, Medical University of Vienna, 1090 Vienna, Austria; felix.eibensteiner@meduniwien.ac.at

**Keywords:** preeclampsia, intrauterine growth restriction, air displacement plethysmography, body composition, fat-free mass, nutrition, preterm infant

## Abstract

Background: Preeclampsia is associated with intrauterine growth restriction (IUGR), which can lead to impaired postnatal growth and neurodevelopment in preterm infants. Preeclampsia can also occur without IUGR and its impact on postnatal nutrition, growth, and body composition remains not fully investigated to the best of our knowledge. Methods: This study included infants born before 37 weeks of gestation who underwent air displacement plethysmography to measure body composition (fat-free mass [FFM] and fat mass [FM]) at term-equivalent age. We compared infants born to mothers with preeclampsia and IUGR (PE-IUGR group) and preeclampsia without IUGR (PE-non-IUGR group) to those born to mothers without preeclampsia (control group). Results: In total, 291 infants were enrolled (control: *n* = 227; PE-non-IUGR: *n* = 43; PE-IUGR: *n* = 21). FFM was significantly lower in the PE-IUGR (mean differences −231 g (IQR: (−373, −88); *p* < 0.001)) and PE-non-IUGR groups (mean differences −260 g (IQR: (−372, −149); *p* < 0.001)) in comparison to the control group. FM was not significantly different between the three groups. Conclusions: This study indicates that infants of preeclamptic mothers, even without IUGR, had significantly lower FFM at term-equivalent age compared to the control group. Further research is necessary to determine if these variations can be modified.

## 1. Introduction

Hypertensive disorders of pregnancy (HDP) occur in 5–10% of all pregnant women worldwide and are leading causes of maternal and fetal morbidity and mortality [1,2,3]. HDP is allocated in the following categories: gestational hypertension, chronic hypertension, preeclampsia, and preeclampsia superimposed upon chronic hypertension [4,5] Preeclampsia is a hypertensive disorder that typically involves the de novo onset of hypertension, including proteinuria, maternal organ failure, and/or uteroplacental dysfunction beyond 20 weeks of pregnancy [4]. A crucial differentiation in the diagnosis of preeclampsia is between early-onset preeclampsia (before 34 weeks of gestation) and late-onset preeclampsia (at or after 34 weeks of gestation) [6]. While there is some overlap in clinical features, early-onset preeclampsia typically presents with severe placental, maternal, and fetal clinical manifestations, resulting in adverse outcomes [7]. Despite ongoing research, there is still no consensus on the origin and pathogenesis of preeclampsia, and its pathophysiology remains not fully understood [8,9,10].

Preeclampsia is the most common serious form of hypertensive pregnancy complications, is linked to increased morbidity and mortality, and leads to a significant number of preterm birth and neonatal morbidity [7]. The long-term impact of preeclampsia impact both women and their newborns [6,7]. Preeclampsia frequently leads to preterm delivery, uteroplacental dysfunction, and prenatal growth restriction with increased rates of complications, like pulmonary hypertension, necrotizing enterocolitis, intraventricular hemorrhages, and, often, mortality, due to the combination of prematurity and extremely low birth weight [11,12]. However, preeclampsia can also occur without fetal growth restriction and with normal fetal doppler ultrasound parameters typically at later gestational ages after 34 weeks of gestation [13]. The etiology and pathomechanism in these cases are not fully investigated [14,15]. Furthermore, the long-term effect on the infant’s health, especially concerning extrauterine growth and body composition, are unclear. Intrauterine growth restriction (IUGR) caused by preeclampsia is correlated with increased fetal and neonatal morbidity and mortality [16]. Ideal postnatal nutrition management and growth assessments for these infants have not been established. Postnatal growth faltering may be best defined by changes in z-scores [17]. However, enhanced neonatal growth, particularly gains in fat-free mass (FFM) measured by body composition, has been associated with brain size and improved neurodevelopment [18,19]. Furthermore, FFM is a good parameter to assess the nutritional status, whereas increased fat mass (FM) is connected with obesity and cardiovascular disorders [11,12].

As far as we know, the effects of preterm infants born to mothers with preeclampsia, particularly those without IUGR, on body composition remain unexplored. Consequently, this study aimed to assess how preeclampsia, both with and without IUGR, influences growth and body composition, particularly focusing on FFM at term-equivalent age.

## 2. Materials and Methods

### 2.1. Study Design and Setting

We conducted a retrospective cohort study that took place at the Department of Pediatrics and Adolescent Medicine, Division for Neonatology and Department of Obstetrics and Gynecology at the Medical University Vienna, Austria. Ethical approval was granted by the local Ethics Committee (Approval Number: 1602/2019).

The main objective of the study was to assess the impact of preeclampsia on growth and body composition in preterm infants. Various growth parameters, such as weight, length, and head circumference at both birth and term-equivalent age, were evaluated. Infants born to mothers with preeclampsia (PE group) were compared with those born to mothers without preeclampsia (control group). Additionally, the preeclampsia group was further subdivided into infants without intrauterine growth restriction (PE-non-IUGR group) and those with intrauterine growth restriction (PE-IUGR group). The definitions of these study groups are described below.

### 2.2. Population

This study comprised all preterm infants born before the 37th week of gestation who were admitted to the hospital between 2017 and 2023 and who had their body composition assessed. According to the local standard protocol, body composition measurements are routinely conducted for infants born preterm at term-equivalent age. Exclusion criteria included chromosomal abnormalities, as well as genetic and metabolic disorders.

Group assignment (control and PE groups) was based on the presence or absence of preeclampsia in the child’s mother. Preeclampsia is determined as de novo onset of hypertension >20th week of gestation and at least one of the following conditions: proteinuria (≥300 mg/day), organ failure (renal dysfunction, hematological complications, such as thrombocytopenia, liver involvement, and neurological complications) or uteroplacental dysfunction, including growth restriction [6,20]. The PE group was further categorized into two subgroups: the PE-non-IUGR group and the PE-IUGR group. Allocation was based on the following criteria: fetal weight below the 10th percentile and evidence of placental insufficiency [21,22]. The diagnosis of IUGR was based on Gordjin et al. [21], which includes an estimated fetal weight < 10th percentile, a uterine artery and/or umbilical artery pulsatility index > 95th percentile, and/or a middle cerebral artery pulsatility index < 5th percentile. The three studies groups were defined in detail as follows: (1) The PE-non-IUGR group was defined as including infants from mothers with preeclampsia with normal Doppler ultrasound measurement and a fetal weight ≥ 10th percentile. (2) The PE-IUGR group was defined as including infants from mothers with preeclampsia with abnormal Doppler ultrasound measurement and a fetal weight < 10th percentile. (3) The control group was defined as infants born to mothers without preeclampsia and a birthweight appropriate for gestational age (between the ≥10th percentile and ≤90th percentile) [21].

Neonatal morbidity was defined as follows: Retinopathy of prematurity (ROP) [23], and intraventricular hemorrhage (IVH) characterized according to the criteria established by Papile et al. [24]. Bronchopulmonary dysplasia (BPD) has been identified as an oxygen demand > 21% at 36 plus 0 weeks of gestation [25]. Necrotizing enterocolitis (NEC) was defined according to the guidelines of Bell et al. [26]. Culture-proven sepsis was identified as a positive bacterial or fungal blood infection, accompanied by symptoms of infection, or antibiotic therapy for >5 days.

Population-based data comprised infants’ age at delivery, sex, antenatal steroid drugs, preterm premature rupture of membranes (PPROM), pathological CTG (cardiotocography), prenatal infection-related preterm delivery and preeclampsia-related preterm delivery, mode of childbirth, APGAR score (5 as well as 10 min), umbilical artery pH, and birth weight, length, and head circumference measurements. Antihypertensive drugs including alpha methyldopa, urapidil, calcium channel blocker, and beta receptor blocker were analyzed. Alpha methyldopa is the local first-line therapy, and the following drugs were also used according to clinical condition and international guidelines: urapidil, calcium channel blocker, and beta receptor blocker [27].

### 2.3. Measurements and Nutrition

Weight, length, and head circumference measurements were conducted at birth, at discharge, and during the clinic follow-up visit when body composition was assessed. Daily weight was assessed and recorded every 48 h once the infant reached 1000 g. Body length was evaluated by a length panel, and head circumference by a flexible ruler.

Body composition was evaluated using air-displacement plethysmography (PEA POD^®^ device; COSMED, Concord, CA, USA). The examination takes 5–7 min and measures fat-free mass (FFM) and fat mass (FM) [5]. It is a simple, reliable and, as it does not require anesthesia, is a safe method of measuring body composition [28,29].

FFM and FM variables were transformed into sex- and age-adjusted Z-Scores based on published reference data [29]. Fenton [30] and WHO growth charts were used for anthropometric data [31]. Anthropometric data were evaluated from childbirth to term age. Weight growth velocity (gram per kg per day) was calculated between day 7 and day 28 (average 2-point method).

Information on the type of feeding was collected during the inpatient stay. Early neonatal feeding began with breastmilk or, if unavailable, pasteurized preterm single donor milk (holder pasteurization) or nutrient-enriched formula for infants born preterm older than 32 weeks of gestation. This feeding regimen was implemented directly after birth and gradually increased between 20–30 milliliter per kg and day. Fortification was started at 100 milliliter per kg and day with Aptamil FMS, Nutricia, Frankfurt, Germany (infants with a gestational age > 26 weeks) or Humavant plus 6, Prolacta Bioscience, California, United States of America (infants < 26 weeks). In addition, parenteral nutrition was imitated at birth, following the ESPGHAN guidelines for carbohydrate, protein, and fat intake [32]. Parenteral nutrition was stopped at 140–160 mL/kg and the day of enteral intake. At discharge, the enteral diet (including breastmilk, fortification, formula, and mixed feeds) was documented.

### 2.4. Statistics

Data analysis was conducted with SPSS version 28 (IBM, New York, NY, USA). A significance level of *p* < 0.05 was applied. Differences in baseline characteristics, growth velocity (measured in grams per kilogram per day from day 7 to day 28), nutrition at discharge (exclusively mother’s own milk), and ROP, BPD, IVH, NEC, and sepsis were compared using Mann–Whitney U or Pearson’s chi-square tests. The Mann–Whitney U and Pearson’s chi-square tests were used accordantly, comparing the expected independent study groups and outcome parameters with the assumption that the data were not normal distributed.

Demographic details were displayed by frequency distribution, median, and interquartile range (IQR). To standardize measurements, anthropometric data and FFM/FM Z-scores were determined using growth charts adjusted for age and sex [33]. Multivariable regression analysis was applied to examine the relationship between FFM and FM. The FFM index (FFMI) and FM index (FMI) were calculated as follows: FM and FFM/lenght2 (kg/m^2^), and weight at term age in the three study groups (control, PE-non-IUGR, PE-IUGR). The model was adjusted for confounding factors, namely sex [34], age at birth, and age at body composition measurement [35]. 

## 3. Results

In this investigation, 291 infants were analyzed (control: *n* = 227; PE: *n* = 64 [non-IUGR: *n* = 43, IUGR: *n* = 21]). Initially, 334 preterm infants were included, although 43 infants in the control group were rejected due to the following factors: large for gestational age: *n* = 6; SGA: *n* = 34; no body composition measurement due to continuous oxygen requirements: *n* = 1; and incomplete follow-up: *n* = 2 (Figure 1).

Baseline characteristics are presented in Table 1. At birth, median gestational age was not significantly different between the study groups (*p* = 0.36). Study groups did not differ significantly in proportion of females or males (control group: 59% male, PE-non-UGR group: 51% male, PE-IUGR group: 48% male) (*p* = 0.86). Anthropometric data were significantly lower at birth in infants in the PE groups compared to the control group (*p* < 0.001), (Table 1). 

Causes of preterm delivery were significantly different between the groups (Table 1). Infection-related preterm delivery was the main reason in the control group and preeclampsia-related preterm delivery was the main reason in the PE groups. Antihypertensive drug therapies in the PE groups are displayed in Table 1. Preeclampsia occurred at a median gestational age of 22.4 weeks (IQR: 20.4; 24.6) in the PE-non-IUGR group and 22.3 weeks (IQR: 20.3; 24.1) in the PE-IUGR group (*p* = 0.68). APGAR score at 10 min and umbilical artery pH were significantly lower in the PE groups compared to the control group (Table 1).

Short-term outcome parameters and nutrition at discharge are detailed in Table 2. Outcome parameters did not show significant differences among the study groups (Table 2). At discharge, infants in all three study groups received primarily their mother’s own milk, and no statistically significant differences were found between the nutritional diet (control vs. PE-non-IUGR, *p* = 0.10, and control vs. PE-IUGR, *p* = 0.27).

Table 3 presents the non-adjusted anthropometric data and FFM as well as FM at term age. 

Median gestational age at the time of body composition measurements showed no significant differences between the groups: control 42.1 weeks (IQR: 40.1; 46.3), PE-IUGR 41.0 weeks (IQR: 39.0; 44.6), and PE-non-IUGR 41.0 weeks (40.0; 44.6); control versus PE-IUGR (*p* = 0.42) and control versus PE-non-IUGR (*p* = 0.21). At term, weight was significantly lower in the PE groups compared to the control group. Both FFM and FM grams were also lower in the PE groups versus controls. The other growth parameters were very similar across groups.

Growth parameters are presented in Table 4. At discharge, median gestational age was not significantly different between the groups: control 38.1 weeks (IQR: 37.0; 40.0), PE-IUGR 38.7 weeks (IQR: 37.4; 39.3) and PE-non-IUGR 38.0 weeks (IQR: 37.0; 39.4); control versus PE-IUGR (*p* = 0.75) and control versus PE-non-IUGR (*p* = 0.29).

Upon discharge, the PE-IUGR group had a significantly lower weight in comparison to the control group (*p* = 0.002). No significant weight difference was observed in the PE- non-IUGR versus controls (*p* = 0.21). In addition, the PE-non-IUGR group had a significantly greater length at discharge than the control group (*p* = 0.011). Length in infants in the PE-IUGR was very similar in comparison to infants in the control groups (*p* = 0.08). Weight velocity did not differ between the PE groups and control groups.

Regression analysis revealed that the primary outcome parameters, FFM Z-score and FFM grams, were significantly lower in infants in the PE-IUGR group compared to the control group (*p* = 0.008 and *p* < 0.01, respectively), and in infants in the PE-non-IUGR group than in the control group (*p* = 0.002 and *p* < 0.001, respectively) (Table 5). The FM Z-score was not significantly different between all study groups. FFMI was significantly lower in the PE-IUGR- and PE-non-IUGR groups (*p* = 0.002, *p* < 0.001, respectively) in comparison to the control group. FMI was not significantly different between the PE groups and the control group (Table 5). Weight at scan was significantly lower in the PE groups in comparison to the control group: control versus PE-IUGR (*p* = 0.003) and control versus PE-non-IUGR (*p* < 0.001) (Table 5).

## 4. Discussion

The study demonstrated that preterm infants born to mothers with preeclampsia had significantly different body composition at term age, particularly a reduction in FFM compared to infants born to mothers without preeclampsia. Furthermore, in a subgroup analysis, we found that infants from preeclamptic mothers without growth restriction during pregnancy had significantly lower FFM at term age than infants in the control group. This study highlights that preeclampsia affects body composition independently of IUGR. Therefore, the presence of preeclampsia should be particularly considered in the postnatal nutritional management of these infants. Emphasis should be placed on individualized postnatal nutritional strategies to address these specific growth and developmental challenges to avoid growth and long-term neurological impairment.

Preeclampsia is a significant cause of preterm birth and is also linked to uteroplacental dysfunction, which is linked to IUGR [6]. Consequently, women with preeclampsia are at higher risk for delivering infants with a low birth weight [36]. Early diagnosis, consistent fetal monitoring, and optimal postnatal management are essential to reduce neonatal and perinatal mortality and morbidity [12]. However, adequate postnatal nutritional management is among the most important preventive interventions [37,38]. Studies have shown that improved nutrition in these infants is essential for optimizing brain size and neurodevelopment [18,39]. In a previous study, we found that standard nutritional management according to ESPGHAN recommendations was insufficient for infants with intrauterine growth retardation [40]. Therefore, individualized nutritional strategies should be considered in these infants.

Growth is generally evaluated including weight, length, and head circumference; however, several studies have demonstrated that assessing qualitative growth by measuring body composition is more accurate for evaluating nutritional management and long-term outcomes [18,41]. In particular, optimizing FFM gain is important for reducing long-term cardiovascular comorbidities [42]. However, preeclampsia can also occur without IUGR, and the possible negative effect on infants’ health, particularly on growth and body composition, is not well understood [14,43]. These infants typically receive standard nutritional management, but preeclampsia without IUGR may still negatively impact uteroplacental nutrient metabolism and extrauterine growth [15,44]. Consequently, the objective of the research was to investigate whether preeclampsia, without IUGR, has an effect on postnatal growth and body composition.

Our investigation revealed that infants in the PE groups had significantly lower body weight at birth and at term compared to control infants. Furthermore, FFM at term-equivalent age was significantly lower in preterm infants in the PE groups, regardless of the presence of IUGR. It is widely recognized that IUGR is linked to compromised postnatal growth, and our data are consistent with previous studies demonstrating that preeclampsia with IUGR negatively affects postnatal growth [45,46]. We also found that preeclampsia without IUGR impacts infant growth and body composition. The data are new and underline the hypothesis that preeclampsia without IUGR has an effect on postnatal growth. We hypothesized that preeclampsia without IUGR and doppler abnormalities might affect placental function and growth factors. In general, preterm infants experience an early separation from the placenta, leading to a premature disruption of growth factors (GF) and hormones [47,48]. The absence of these factors, such as placental growth hormone, human placental lactogen, maternal insulin-like GF 1 and 2, corticotropin-releasing hormone, leptin, insulin and thyroid hormones, can result in growth faltering and suboptimal body composition [48]. The metabolic endocrine disorder associated with preeclampsia is often unrecognized. Supplementation of the growth-stimulating hormones, such as insulin-like GF 1 and thyroid hormones, in infants born preterm, may help to support reduced growth faltering [48]. However, preeclampsia is additionally linked to a reduction in growth factors and hypoalbuminemia, both of which are associated with poor growth and may negatively affect extrauterine growth in these infants [49,50,51]. Furthermore, Roberts et al. [52] have shown that vascular remodeling in the arteries of women with PE is substantially different from in those without PE. These factors can substantially influence postnatal growth and neurodevelopment. Additionally, infant nutrition, particularly the composition of breast milk in mothers with preeclampsia, may affect postnatal growth. Previous studies [53] have demonstrated that lipid metabolism and lactogenesis are impaired in women with preeclampsia, which could influence breast milk composition and infant growth. An exploratory study by Beser et al. [54] showed that macronutrients in colostrum were not affected by preeclampsia. However, the analysis of macronutrients at different stages of lactation has not yet been investigated, and research with a larger cohort is needed. Our investigation underlines the hypothesis that these infants are at increased risk for growth faltering, and an adequate nutritional management is essential for this specific patient group. IUGR is associated with poor neurodevelopment, cardiovascular disease, reduced lung function, renal impairment, increased insulin resistance, and metabolic syndrome [55,56,57]. Studies have demonstrated that high FM, as measured by body composition, is linked to being overweight and cardiovascular diseases [47,57]. We also assessed FM in the study groups but failed to show differences in FM Z-scores between the PE groups and controls. This indicates that the PE groups might not be at an increased risk of developing obesity later in life.

After birth, nutritional management is particularly challenging for growth-retarded children. Infants with IUGR frequently need extended periods of parenteral nutrition, as establishing enteral nutrition is slower because of restricted food tolerance and reduced nutrient reserves and risk of complications, such as necrotizing enterocolitis [58]. Early aggressive nutritional management is attempted to counteract appropriate growth [59]. Preterm infants should receive fortified breastmilk or formula at least until term-equivalent age to ensure optimal growth [32]. Based on recommendation by European Society for Pediatric Gastroenterology, Hepatology, and Nutrition (ESPGHAN), fortification is still suggested at discharge, if an infant does not exceed the 10th percentile [32]. In our study, nutrition was managed according to ESPGHAN guidelines but was not adequate to prevent FFM loss in preterm infants in the PE groups, regardless of whether or not the fetus as growth-restricted.

These data highlight the necessity of early individualized nutritional management in infants born from preeclamptic mothers with intrauterine growth restriction to prevent malnutrition and poor postnatal growth [17]. Overall, breast milk is the ideal feeding option for these infants because of its positive impact on cardiovascular, neurological, and growth outcomes. Pasteurized donor milk is the best alternative choice [42]. In recent years, attention has focused on optimizing postnatal growth through nutritional interventions [60]. A study by Perrin et al. [61] showed that premature infants benefit from an individually adapted diet. Targeted fortification to optimize protein and energy content would be particularly beneficial for growth-retarded infants [62,63]. However, analyzing breast milk or human milk is very time-consuming and resource-demanding [61]. Further studies and randomized controlled trials are necessary to assess individualized nutritional management for these infants. 

A key strength of this research is the qualitative monitoring of growth by body composition measurement, utilizing not only traditional anthropometric parameters, such as body weight, but also others. While weight gain and length growth alone are not optimal indicators of nutritional status [47], body composition measurements provide both quantitative and qualitative information on growth, distinguishing between FM and FFM [64]. Monitoring body composition is, thus, a fundamental part of improving nutritional outcomes [18,41,65]. FFM, in particular, is associated with brain size and serves as a good marker of neurodevelopmental outcomes [18,41]. Consequently, routine measurements could be a good and easy method to assess the nutritional status as well as subsequent neurodevelopment. Furthermore, it is an easy and safe approach for investigating adequate growth in preterm infants. Further studies are required to investigate future clinical and research implications. 

Two weaknesses of the current study are the comparatively small cohort and its retrospective character. However, the number of infants in the PE group (*n* = 64) is relatively large for this population, providing new insights into the effects of preeclampsia without IUGR on infants’ growth and body composition. Further research is needed to improve insight in the pathomechanism and long-term effects of preeclampsia without IUGR on the health of these preterm infants.

Our study found that FFM is significantly lower in the PE group, even in infants born to mothers with preeclampsia without IUGR. In comparison to our previous study, the body composition measurements of the PE groups showed very similar FFM Z-scores (PE-non-IUGR FFM Z-score −1.6 and PE-IUGR FFM Z-score −1.5) [40]. The FFM Z-scores of the control group in our previous and the current studies were consistent (FFM Z-score −1.1) [40]. Additionally, FFMI was calculated and found to be significantly lower in the PE groups compared to the control group. These data emphasize the importance of individualized and enhanced nutrition in these infants to improve long-term health.

## 5. Conclusions

A significant decrease in FFM was observed in infants born to mothers with preeclampsia compared to those born to mothers without preeclampsia. Even infants born to preeclamptic mothers who did not experience growth restriction during pregnancy had significantly lower FFM and body weight at term-equivalent age. This finding suggests that infants born to mothers with preeclampsia have altered body composition. Therefore, research is needed to understand whether these differences are modifiable.

## Figures and Tables

**Figure 1 nutrients-16-03627-f001:**
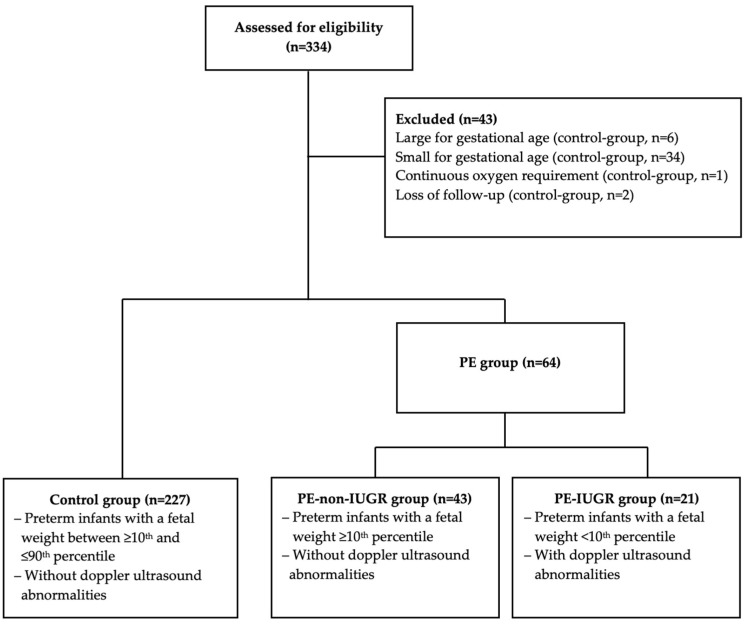
Overview of the study groups.

**Table 1 nutrients-16-03627-t001:** Baseline characteristics.

Variables	Control Group (AGA-Group) (*n* = 227)	PE-Non-IUGR Group (*n* = 43)	PE IUGR Group (*n* = 21)	*p*-Values
Gestational age, weeks *	26.4 (25.0; 28.0)	27.6 (23.6; 32.1)	26.6 (25.1; 28.2)	0.36
Male, % (n)	59 (133/227)	51 (22/43)	48 (10/21)	0.86
Antenatal steroids, % (n)	84 (190/227)	88 (38/43)	86 (18/21)	0.73
PPROM, % (n)	46 (104/227)	7 (3/43)	5 (1/21)	<0.001
Pathological CTG, % (n)	17 (39/227)	28 (12/43)	38 (8/21)	0.03
Infection-related delivery, % (n)	73 (165/227)	7 (3/43)	5 (1/21)	<0.001
PE-related delivery, % (n)	0 (0/0)	93 (40/43)	95 (20/21)	<0.001
Antihypertensive drugs, % (n)				
- Alpha methyldopa	0 (0/0)	100 (43/43)	100 (21/21)	<0.001
- Drug combination **	0 (0/0)	23 (10/43)	28 (6/21)	<0.001
Caesarean delivery, % (n)	68 (155/227)	70 (30/43)	71 (15/21)	0.005
APGAR Score, 5 min *	9 (8, 9)	9 (8, 9)	9 (8/9)	0.40
APGAR Score, 10 min *	9 (9, 9)	9 (9, 9)	9 (9/9)	0.019
Umbilical artery, pH *	7.34 (7.28, 7.38)	7.31 (7.29, 7.33)	7.30 (7.27, 7.32)	0.024
Birth weight, gram *	910 (740, 1185)	820 (630, 1522)	600 (505, 850)	<0.001
Birth weight, Z-Score *	0.1 (−0.4, 0.5)	−0.7 (−1.1, −0.8)	−1.5 (−1.6, −1.3)	<0.001
Birth length, cm *	35 (32, 38)	35 (31, 41)	30 (29, 35)	<0.001
Birth length, Z-Score *	0.1 (−0.5, 0.8)	−0.5 (−0.9, −0.2)	−1.5 (−1.7, −1.4)	<0.001
Birth HC, cm *	25.0 (23.0, 26.5)	25.0 (21.0, 28.0)	22.0 (21.2, 25.2)	<0.001
Birth HC, Z-Score *	0.3 (−0.3, 1.0)	−0.5 (−0.8, −0.3)	−1.5 (−1.7, −1.4)	<0.001

* Data presented in median (interquartile range); preeclampsia (PE), preterm premature rupture of the membrane. (PPROM), head circumference (HC), cardiotocography (CTG). ** Additional antihypertensive drug therapy: urapidil, calcium channel blocker, or beta receptor blocker.

**Table 2 nutrients-16-03627-t002:** Outcome parameters and nutrition at discharge.

	Control Group (*n* = 227)	PE-Non-IUGR Group (*n* = 43)	PE-IUGR Group (*n* = 21)	Control vs. Non-IUGR *p*-Values	Control vs. IUGR *p*-Values
Neonatal morbidities					
IVH (stage ≥ 3), % (n)	8 (19/227)	5 (2/43)	5 (1/21)	0.60	0.19
ROP (stage ≥ 3), % (n)	14 (31/227)	9 (4/43)	14 (3/21)	0.90	0.30
BPD, % (n)	11 (25/227)	9 (4/43)	10 (2/21)	0.17	0.27
NEC (stage ≥ 2), % (n)	7 (16/227)	5 (2/43)	5 (1/21)	0.60	0.17
Culture proven sepsis, % (n)	22 (49/227)	14 (6/43)	19 (4/21)	0.28	0.09
Nutrition at discharge					
Exclusive mother’s own milk at discharge, % (n)	54 (122/227)	67 (29/43)	67 (14/21)	0.10	0.27

Values are median (interquartile range); intraventricular hemorrhage (IVH), retinopathy of prematurity (ROP), bronchopulmonary dysplasia (BPD), and necrotizing enterocolitis (NEC).

**Table 3 nutrients-16-03627-t003:** Non-adjusted anthropometric parameters and body composition measurements.

Variables	Control Group (*n* = 227)	PE-Non-IUGR Group (*n* = 43)	PE-IUGR Group (*n* = 21)
Age at measurement, week	42.1 (40.1, 46.3)	41.0 (40.0, 446)	41.0 (39.0, 44.6)
Anthropometric parameters at term-equivalent age *			
Weight, gram	3590 (2944, 4470)	3168 (2731, 4272)	3024 (2605, 3650)
Length, cm	51.0 (49.0, 55.0)	53.0 (51.0, 55.0)	51.0 (48.0, 54.0)
Head circumference, cm	35.0 (34.0, 37.5)	37.5 (35.0, 38.2)	36.0 (33.0, 38.0)
Body composition parameters at term-equivalent age *			
FFM, percentage	78.3 (73.9, 83.6)	79.0 (74.3, 81.0)	78.5 (73.3, 82.6)
FM, percentage	21.7 (16.5, 26.1)	21.0 (19.0, 25.7)	21.5 (17.4, 26.7)
FFM, gram	2821 (2440, 3323)	2527 (2195, 3173)	2425 (2144, 2631)
FM, gram	769 (504, 1147)	641 (536, 1099)	599 (461, 1019)

* Data displayed are median (interquartile range).

**Table 4 nutrients-16-03627-t004:** Growth parameters.

Variables	Control Group (*n* = 227)	PE-Non-IUGR Group (*n* = 43)	PE-IUGR Group (*n* = 21)	Control vs. Non-IUGR *p*-Values	Control vs. IUGR *p*-Values
Growth velocity from day 7 to day 28 *					
Age at discharge, week	38.1 (37.0, 40.0)	38.0 (37.0, 39.4)	38.7 (37.4, 39.3)	0.29	0.75
Discharge weight, gram	2785 (2433, 3075)	2750 (2425, 2970)	2462 (2150, 2790)	0.21	0.002
Weight velocity, g/kg/d	15.2 (12.5, 17.1)	14.6 (11.6, 16.3)	14.4 (11.9, 16.5)	0.26	0.24
Discharge length, cm	46.0 (44.0, 48.0)	47.0 (46.0, 49.0)	46.0 (44.5, 46.0)	0.011	0.08
Discharge HC, cm	32.6 (31.5, 33.9)	33.0 (32.0, 33.5)	32.0 (31.5, 32.0)	0.80	0.004

* Data shown in median (interquartile range).

**Table 5 nutrients-16-03627-t005:** Weight and fat mass and fat-free mass data for control, PE-non-IUGR and PE-IUGR groups.

	Adjusted Mean			Adjusted Mean Difference	
	Control Group	PE-Non-IUGR Group	PE-IUGR Group	PE-Non-IUGR Group	PE-IUGR Group
Total (n)	227	43	21		
Weight at scan, gram ^1^	3836 (3775, 3897)	3483 (3337, 3630)	3519 (3336, 3702)	−353 (−512, −193) *p* < 0.001	−317 (−508, −126) *p* = 0.003
FFM, Z-score	−1.0 (−1.2, −0.9)	−1.6 (−1.9, −1.4)	−1.5 (−1.8, −1.2)	−0.6 (−0.8, −0.5) *p* = 0.002	−0.5 (−0.7, −0.3) *p* = 0.008
FM, Z-score	1.0 (0.9, 1.1)	0.7 (0.4, 1.0)	0.7 (0.3, 1.1)	0.2 (0.1, 0.3) *p* = 0.24	0.1 (−0.1, 0.2) *p* = 0.41
FFM, gram	2959 (2919, 3000)	2699 (2597, 2801)	2728 (2592, 2866)	−260 (−372, −149) *p* < 0.001	−231 (−373, −88) *p* < 0.001
FM, gram	864 (824, 904)	781 (686, 876)	789 (660, 917)	83 (−21, 187) *p* = 0.117	75 (−75, 207) *p* = 0.334
FFM, Index	10.8 (10.6, 10.9)	9.9 (9.5, 10.3)	10.0 (9.5, 10.5)	−0.9 (−1.3, −0.6) *p* < 0.001	−0.8 (−1.4, −0.3) *p* = 0.002
FM, Index	3.0 (2.8, 3.1)	2.9 (2.6, 3.3)	2.7 (2.2, 3.2)	−0.1 (−0.4, 0.3) *p* = 0.80	−0.3 (−0.8, 1.5) *p* = 0.18

^1^ Mean (95% CI) adjusted for sex, postmenstrual age at measurement, and age at birth.

## Data Availability

Data used in the study are available on request (corresponding author). The data are not publicly available because of studies in progress.

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
