# Peer review of "Preeclampsia and Future Implications on Growth and Body Composition in Preterm Infants"

_nutrients, 2024, doi:10.3390/nu16213627_

Round 1

Reviewer 1 Report

Comments and Suggestions for Authors

The manuscript presented to me for the price takes on a very interesting topic. The authors did an excellent job of presenting the topic, but I miss a few pieces of information in the methodology:

1. What was the cause of premature birth in the control group, whether the women were under medical care, whether they were taking any medications

2. In the study group, there is no information about the time of diagnosis of preeclampsia in women. The authors explained that this condition is most often diagnosed after the 20th week of pregnancy, but at what time was it diagnosed in the women studied?

3. Were the women in the study group treated, and if so, how

4. What was the weight gain of the women in both groups throughout the pregnancy

5. In which week of pregnancy, if they were premature babies, were the children born?

Reviewer 2 Report

Comments and Suggestions for Authors

The work provides a comprehensive overview of preeclampsia, but it could benefit from clearer differentiation between the types of preeclampsia (with and without IUGR). Expanding on the specific criteria used to classify patients into these groups would enhance clarity, especially since the distinction is critical for the study's conclusions.

While the suthors mention the use of Mann-Whitney-U and Pearson-Chi-Square tests, a more detailed explanation of why these specific tests were chosen for comparing growth velocity and neonatal morbidity would be helpful for the reader. Including an explanation of how assumptions of the statistical tests were verified would strengthen the methodological transparency.

The manuscript presents strong findings on the association between fat-free mass (FFM) and neurodevelopmental outcomes, but it would be beneficial to explore this in more depth. Incorporating a section on how these findings could translate into long-term monitoring or interventions for preterm infants born to preeclamptic mothers would provide additional clinical relevance.

Reviewer 3 Report

Comments and Suggestions for Authors

This paper addresses an interesting question: “the aim of this study was to evaluate the effect of preeclampsia with and without IUGR on growth and body composition, especially FFM at term-equivalent age” but it needs considerable simplification and shortening since the study and its findings get lost in the many sentences that overstate the evidence. Importantly, the paper is not at all clear about its exposures and outcomes. 

The primary problem is the need to clarify whether the groups were categorized using the Gordijn definition of IUGR or birthweight percentiles. The study exposure is preeclampsia +/- /IUGR or SGA. The paper refers to the Gordijn definition of IUGR but then the flow diagram only separates the pre-eclampsia groups by birthweight. Also the cut point between the IUGR and AGA groups are not clear about where infants who had a birthweight at the 10th percentile would fall. When birthweight is used the tradition is to consider greater than or equal to the 10th percentile as not SGA and those less than the 10th percentile as SGA. Then the naming of the groups need to use either IUGR and non-IUGR or SGA and AGA based on the actual categorizations used. 

The paper and abstract should be written more efficiently to describe the three exposure groups throughout the paper and abstract. For example, Table 1 needs to be 3 groups and these 2 sentences in the abstract need to be written more efficiently: “FFM was

significantly lower in infants born to mothers with preeclampsia, whereas FM

was not significantly different between the PE group and the control group.

Infants born to preeclamptic mothers without IUGR during pregnancy had also

significantly lower FFM at term-equivalent age than infants in the control

group”.

What body composition reference values  were used to obtain z-scores for the primary outcome for group comparisons? The paper says “FFM and FM variables were transformed into sex- and age-adjusted Z-Scores based on published reference charts [29]”  but reference 29 does not have reference values and it concludes: “none of the currently described methods give an accurate and practically achievable method of obtaining body composition measures in preterm infants in day to day routine clinical practise”. 

It is best to omit “extrauterine growth restriction” and “EUGR” from the abstract and paper for four reasons: 

1.     The following sentence “Following birth, premature infants, particularly those with IUGR, face a heightened susceptibility to extrauterine growth retardation (EUGR) [17]” reflects unrealistic expectations as neither ESPGHAN nor the American Academy of Pediatrics recommend that SGA infants grow faster to cross percentiles upward, which would be needed for those infants to not be <10thpercentile later. Also EUGR has not been found to be predictive of cognitive impairment nor neurodevelopmental outcomes. A better growth measure is changes in z-scores as a continuous measure or >2 z-scores as reflected in these studies Shah 2006 and El Rafei 2021/3.

2.     Experts recommend not using the phrase EUGR since it is a misnomer.

3.     Your references 16, 36 and 38 are either not about EUGR and/or do not justify using it.

4.     “The extrauterine growth of premature infants is inferior to intrauterine growth in terms of both rate and quality” needs to be omitted as it is an over generalization and not likely true since it is based on an over concern about infants placing low on growth charts, which some of can be explained by the postnatal weight loss. 

It is not appropriate to claim that a topic “has not been investigated so far” since it may be been and is not yet published or seen. The words “To the best of our knowledge” as you have in the text is one way to make this statement valid.

Comments on the Quality of English Language

The problems are described for the authors. I did not see English language problems

Round 2

Reviewer 1 Report

Comments and Suggestions for Authors

The authors did not respond to my comments.

Reviewer 3 Report

Comments and Suggestions for Authors

Thank you for the improvements to your manuscripts, which have greatly improved it.

There’s an error in your abstract, where you repeat the numbers for fat-free mass differences. 

It is not appropriate to adjust for current weight and length, because doing so over adjusts in the regression model . References:

a. Kramer MS, Zhang X, Dahhou M, Yang S, Martin RM, Oken E, et al. Does fetal growth restriction cause later obesity? Pitfalls in analyzing causal mediators as confounders. Am J Epidemiol. 2017 Apr;185(7):585–90. 

b. Ananth C V., Schisterman EF. Confounding, causality, and confusion: the role of intermediate variables in interpreting observational studies in obstetrics. Am J Obstet Gynecol [Internet]. 2017;217(2):167–75. 

c. Elmrayed S, Metcalfe A, Brenner D. Wollny K. & Fenton TR. Are small-for-gestational-age preterm infants at increased risk of overweight? Statistical pitfalls in overadjusting for body size measures. J Perinatol (2021). PMID: 33850286 https://rdcu.be/ciB7k

The weight gain velocity in g/kg/day are unbelievably high numbers for these gestational ages. Here is a reference for recommended calculation methods for growth velocity. (PMID: 30705399)

The sentence in your abstract would be improved if the word significantly is omitted, and if the sentences reworded to: This study observed that preterm infants of mothers with pre-eclampsia, with or without IUGR, had lower FFM at term equivalent age compared to the control group without preeclampsia. This sentence should not refer to impaired neurodevelopment because that was not part of your study and that can be difficult for parents to read. It is also important to realize that while pre-eclampsia is a risk factor for adverse neurodevelopment, which means it has been associated, it does not predict adverse neurodevelopment for all those affected. Please review your paper for sentences that suggest that all might be affected by altered body composition and/or suboptimal neurodevelopment and revise. 

“Especially, the postnatal nutritional management is challenging in these infants and a previous studies showed that postnatal growth failure defined by Z-Score change is a good parameter for long-term neurodevelopment (17).” Would be improved as:

Ideal postnatal nutrition management and growth assessments for these infants have not been established. Postnatal growth faltering may be best defined by changes in z-scores (17).” Experts also recommend using postnatal growth “faltering” instead of postnatal growth “failure”.

Since we do not know how to alter body composition after pre-eclmapsia, please edit: “This finding emphasizes that infants born to mothers with preeclampsia face a higher risk of experiencing EUGR altered body composition. Therefore, implementing individualized nutritional strategies for these infants is crucial to ensure optimal growth, body composition, and neurodevelopment.” To “This finding suggests that infants born to mothers with preeclampsia have altered body composition. Therefore, research is needed to understand whether these differences are modifiable.”

The last sentence of the abstract also needs to be revised in this way.

Please omit this sentence since the one above covers these infants as well: “Our data underline that even infants without intrauterine growth restriction are at higher risk for altered body composition. Nutritional management needs to be adapted for these infants and adequate and individualized nutrition may help to avoid growth failure, which is associated with a delayed neurodevelopment (17).

Comments on the Quality of English Language

Some editing needed

Round 3

Reviewer 3 Report

Comments and Suggestions for Authors

Thank you for the improvements to your manuscripts, which have greatly improved it. 

The infants born to women with pre-eclampsia were smaller and there is a metric that should be included to evaluate their fat mass, that is the fat mass index, which is used in this study: https://pubmed.ncbi.nlm.nih.gov/29551311/. Your manuscript will be strengthened if this FMI and FFMI are added. If this index is not significantly different between any of the groups then it should be the metric featured in your abstract since it takes the differences in weight into account. 

Comments on the Quality of English Language

No concerns
